# Levenshtein Transformer

**Jiatao Gu[†], Changhan Wang[†], and Jake Zhao (Junbo)[‡◇]**
[†]Facebook AI Research
[‡]New York University  [◇]Tigerobo Inc.
[†]{jgu, changhan}@fb.com [‡]jakezhao@cs.nyu.edu

## Abstract

Modern neural sequence generation models are built to either generate tokens step-by-step from scratch or (iteratively) modify a sequence of tokens bounded by a fixed length. In this work, we develop Levenshtein Transformer, a new partially autoregressive model devised for more flexible and amenable sequence generation. Unlike previous approaches, the basic operations of our model are *insertion* and *deletion*. The combination of them facilitates not only generation but also sequence refinement allowing dynamic length changes. We also propose a set of new training techniques dedicated at them, effectively exploiting one as the other's learning signal thanks to their complementary nature. Experiments applying the proposed model achieve comparable or even better performance with much-improved efficiency on both generation (e.g. machine translation, text summarization) and refinement tasks (e.g. automatic post-editing). We further confirm the flexibility of our model by showing a Levenshtein Transformer trained by machine translation can straightforwardly be used for automatic post-editing. [1]

## 1 Introduction

Neural sequence generation models are widely developed and deployed in tasks such as machine translation (Bahdanau et al., 2015; Vaswani et al., 2017). As we examine the current frameworks, the most popular autoregressive models generate tokens step-by-step. If not better, recent non-autoregressive approaches (Gu et al., 2018; Kaiser et al., 2018; Lee et al., 2018) have proved it possible to perform generation within a much smaller number of decoding iterations.

In this paper, we propose Levenshtein Transformer (LevT), aiming to address the lack of flexibility of the current decoding models. Notably, in the existing frameworks, the length of generated sequences is either fixed or monotonically increased as the decoding proceeds. This remains incompatible with human-level intelligence where humans can revise, replace, revoke or delete any part of their generated text. Hence, LevT is proposed to bridge this gap by breaking the in-so-far standardized decoding mechanism and replacing it with two basic operations — *insertion* and *deletion*.

We train the LevT using imitation learning. The resulted model contains two policies and they are executed in an alternate manner. Empirically, we show that LevT achieves comparable or better results than a standard Transformer model on machine translation and summarization, while maintaining the efficiency advantages benefited from parallel decoding similarly to (Lee et al., 2018). With this model, we argue that the decoding becomes more flexible. For example, when the decoder is given an empty token, it falls back to a normal sequence generation model. On the other hand, the decoder acts as a refinement model when the initial state is a low-quality generated sequence. Indeed, we show that a LevT trained from machine translation is directly applicable to translation post-editing without

any change. This would not be possible with any framework in the literature because generation and refinement are treated as two different tasks due to the model's inductive bias.

One crucial component in LevT framework is the learning algorithm. We leverage the characteristics of *insertion* and *deletion* — they are complementary but also adversarial. The algorithm we propose is called "dual policy learning". The idea is that when training one policy (insertion or deletion), we use the output from its adversary at the previous iteration as input. An *expert* policy, on the other hand, is drawn to provide a correction signal. Despite that, in theory, this learning algorithm is applicable to other imitation learning scenarios where a dual adversarial policy exists, in this work we primarily focus on a proof-of-concept of this algorithm landing at training the proposed LevT model.

To this end, we summarize the contributions as follows:

- We propose Levenshtein Transformer (LevT), a new sequence generation model composed of the insertion and deletion operations. This model achieves comparable or even better results than a strong Transformer baseline in both machine translation and text summarization, but with much better efficiency (up to $\times 5$ speed-up in terms of actual machine execution time);

- We propose a corresponding learning algorithm under the theoretical framework of imitation learning, tackling the complementary and adversarial nature of the dual policies;

- We recognize our model as a pioneer attempt to unify sequence generation and refinement, thanks to its built-in flexibility. With this unification, we empirically validate the feasibility of applying a LevT model trained by machine translation directly to translation post-editing, without any change.

## 2 Problem Formulation

### 2.1 Sequence Generation and Refinement

We unify the general problems of sequence generation and refinement by casting them to a Markov Decision Process (MDP) defined by a tuple $(\mathcal{Y}, \mathcal{A}, \mathcal{E}, \mathcal{R}, \boldsymbol{y_0})$. We consider the setup consisting an agent interacting with an environment $\mathcal{E}$ which receives the agent's editing actions and returns the modified sequence. We define $\mathcal{Y} = \mathcal{V}^{N_{\max}}$ as a set of discrete sequences up to length $N_{\max}$ where $\mathcal{V}$ is a vocabulary of symbols. At every decoding iteration, the agent receives an input $\boldsymbol{y}$ drawn from scratch or uncompleted generation, chooses an action $\boldsymbol{a}$ and gets a reward $r$. We use $\mathcal{A}$ to denote the set of actions and $\mathcal{R}$ for the reward function. Generally the reward function $\mathcal{R}$ measures the distance between the generation and the ground-truth sequence, $\mathcal{R}(\boldsymbol{y}) = -\mathcal{D}(\boldsymbol{y}, \boldsymbol{y}^*)$ which can be any distance measurement such as the Levenshtein distance (Levenshtein, 1965). It is crucial to incorporate $\boldsymbol{y_0} \in \mathcal{Y}$ into the our formulation. As the initial sequence, the agent receives—when $\boldsymbol{y_0}$ is an already generated sequence from another system, the agent essentially learns to do refinement while it falls back to generation if $\boldsymbol{y_0}$ is an empty sequence. The agent is modeled by a policy, $\pi$, that maps the current generation over a probability distribution over $\mathcal{A}$. That is, $\pi : \mathcal{Y} \to P(\mathcal{A})$.

### 2.2 Actions: Deletion & Insertion

Following the above MDP formulation, with a subsequence $\boldsymbol{y}^k = (y_1, y_2, ..., y_n)$, the two basic actions – *deletion* and *insertion* – are called to generate $\boldsymbol{y}^{k+1} = \mathcal{E}(\boldsymbol{y}^k, \boldsymbol{a}^{k+1})$. Here we let $y_1$ and $y_n$ be special symbols `<s>` and `</s>`, respectively. Since we mainly focus on the policy of a single round generation, the superscripts are omitted in this section for simplicity. For conditional generation like MT, our policy also includes an input of source information $\boldsymbol{x}$ which is also omitted here.

**Deletion** The deletion policy reads the input sequence $\boldsymbol{y}$, and for every token $y_i \in \boldsymbol{y}$, the deletion policy $\pi^{\mathrm{del}}(d|i, \boldsymbol{y})$ makes a binary decision which is 1 (delete this token) or 0 (keep it). We additionally constrain $\pi^{\mathrm{del}}(0|1, \boldsymbol{y}) = \pi^{\mathrm{del}}(0|n, \boldsymbol{y}) = 1$ to avoid sequence boundary being broken. The deletion classifier can also be seen as a fine-grained discriminator used in GAN (Goodfellow et al., 2014) where we predict "fake" or "real" labels for every predicted token.

**Insertion** In this work, it is slightly more complex to build the insertion atomic because it involves two phases: *placeholder* prediction and *token* prediction so that it is able to insert multiple tokens at the same slot. First, among all the possible inserted slots $(y_i, y_{i+1})$ in $\boldsymbol{y}$, $\pi^{\mathrm{plh}}(p|i, \boldsymbol{y})$ predicts the possibility of adding one or several placeholders. In what follows, for every placeholder predicted as

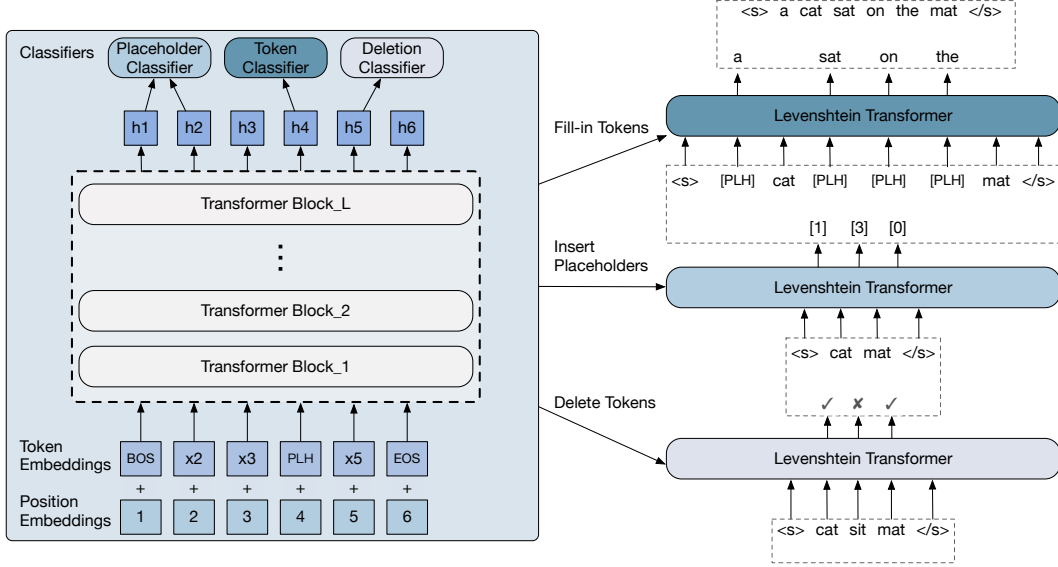

Figure 1: The illustration of the proposed Levenshtein Transformer decoder for **one refinement iteration**. The same architecture can be applied for three different tasks with specific classifiers. For simplicity, the encoder-decoder attention is omitted within each Transformer-Block.

above, a token prediction policy $\pi^{\text{tok}}(t|i, \boldsymbol{y})$ replaces the placeholders with actual tokens in the vocabulary. The two-stage insertion process can be also be viewed as a hybrid of Insertion Transformer (Stern et al., 2019) and masked language model (MLM, Devlin et al., 2018; Ghazvininejad et al., 2019).

**Policy combination** Recall that our two operations are complementary. Hence we combine them in an alternate fashion. For example in sequence generation from the empty, insertion policy is first called and it is followed by deletion, and then repeat till the certain stopping condition is fulfilled. Indeed, it is possible to leverage the parallelism in this combination. We essentially decompose one iteration of our sequence generator into three phases: "delete tokens – insert placeholders – replace placeholders with new tokens". Within each stage, all operations are performed in parallel. More precisely, given the current sequence $\boldsymbol{y} = (y_0, \ldots, y_n)$, and suppose the action to predict is $\boldsymbol{a} = \{\underbrace{d_0, \ldots d_n}_{\boldsymbol{d}}; \underbrace{p_0, \ldots, p_{n-1}}_{\boldsymbol{p}}; \underbrace{t_0^1, \ldots t_0^{p_0}, \ldots, t_{n-1}^{p_{n-1}}}_{\boldsymbol{t}}\}$, the policy for one iteration is:

$$\pi(\boldsymbol{a}|\boldsymbol{y}) = \prod_{d_i \in \boldsymbol{d}} \pi^{\text{del}}(d_i|i, \boldsymbol{y}) \cdot \prod_{p_i \in \boldsymbol{p}} \pi^{\text{plh}}(p_i|i, \boldsymbol{y}') \cdot \prod_{t_i \in \boldsymbol{t}} \pi^{\text{tok}}(t_i|i, \boldsymbol{y}''), \tag{1}$$

where $\boldsymbol{y}' = \mathcal{E}(\boldsymbol{y}, \boldsymbol{d})$ and $\boldsymbol{y}'' = \mathcal{E}(\boldsymbol{y}', \boldsymbol{p})$. We parallelize the computation within each sub-tasks.

## 3 Levenshtein Transformer

In this section, we cover the specs of Levenshtein Transformer and the dual-policy learning algorithm. Overall our model takes a sequence of tokens (or none) as the input then iteratively *modify* it by alternating between insertion and deletion, until the two policies combined converge. We describe the detailed learning and inference algorithms in the Appendix.

### 3.1 Model

We use Transformer (Vaswani et al., 2017) as the basic building block. For conditional generation, the source $\boldsymbol{x}$ is included in each TransformerBlock. The states from the $l$-th block are:

$$\boldsymbol{h}_0^{(l+1)}, \boldsymbol{h}_1^{(l+1)}, ..., \boldsymbol{h}_n^{(l+1)} = \begin{cases} E_{y_0} + P_0, E_{y_1} + P_1, ..., E_{y_n} + P_n, & l = 0 \\ \text{TransformerBlock}_l(\boldsymbol{h}_0^{(l)}, \boldsymbol{h}_1^{(l)}, ..., \boldsymbol{h}_n^{(l)}), & l > 0 \end{cases} \tag{2}$$

where $E \in \mathbb{R}^{|\mathcal{V}| \times d_{\text{model}}}$ and $P \in \mathbb{R}^{N_{\max} \times d_{\text{model}}}$ are the token and position embeddings, respectively. We show the illustration of the proposed LevT model for one refinement (delete, insert) as Figure 1.

**Policy Classifiers** The decoder outputs $(\boldsymbol{h}_0, \boldsymbol{h}_2, ..., \boldsymbol{h}_n)$ are passed to three policy classifiers:

1. *Deletion Classifier*: LevT scans over the input tokens (except for the boundaries) and predict "deleted" (0) or "kept" (1) for each token position,

$$\pi_\theta^{\mathrm{del}}(d|i, \boldsymbol{y}) = \mathrm{softmax}\left(\boldsymbol{h}_i \cdot A^\top\right), \ \ i = 1, \ldots n-1, \tag{3}$$

   where $A \in \mathbb{R}^{2 \times d_{\mathrm{model}}}$, and we always keep the boundary tokens.

2. *Placeholder Classifier*: LevT predicts the number of tokens to be inserted at every consecutive position pairs, by casting the representation to a categorical distribution:

$$\pi_\theta^{\mathrm{plh}}(p|i, \boldsymbol{y}) = \mathrm{softmax}\left(\mathtt{concat}(\boldsymbol{h}_i, \boldsymbol{h}_{i+1}) \cdot B^\top\right), \ \ i = 0, \ldots n-1, \tag{4}$$

   where $B \in \mathbb{R}^{(K_{\max}+1) \times (2d_{\mathrm{model}})}$. Based on the number $(0 \sim K_{\max})$ of tokens it predicts, we insert the considered number of placeholders at the current position. In our implementation, placehoder is represented by a special token <PLH> which was reserved in the vocabulary.

3. *Token Classifier*: following the placeholder prediction, LevT needs to fill in tokens replacing all the placeholders. This is achieved by training a token predictor as follow:

$$\pi_\theta^{\mathrm{tok}}(t|i, \boldsymbol{y}) = \mathrm{softmax}\left(\boldsymbol{h}_i \cdot C^\top\right), \ \ \forall y_i = \texttt{<PLH>}, \tag{5}$$

   where $C \in \mathbb{R}^{|\mathcal{V}| \times d_{\mathrm{model}}}$ with parameters being shared with the embedding matrix.

**Weight Sharing** Our default implementation always assumes the three operations to share the same Transformer backbone to benefit features learned from other operations. However, it is also possible to disable weight sharing and train separate decoders for each operations, which increases the capacity of the model while does not affect the overall inference time.

**Early Exit** Although it is parameter-efficient to share the same Transformer architecture across the above three heads, there is room for improvement as one decoding iteration requires three full passes of the network. To make trade-off between performance and computational cost, we propose to perform *early exit* (attaching the classifier to an intermediate block instead of the last one) for $\pi^{\mathrm{del}}$ and $\pi^{\mathrm{plh}}$ to reduce computation while keeping $\pi^{\mathrm{tok}}$ always based on the last block, considering that token prediction is usually more challenging than the other two tasks.

### 3.2 Dual-policy Learning

**Imitation Learning** We use imitation learning to train the Levenshtein Transformer. Essentially we let the agent imitate the behaviors that we draw from some expert policy $\pi^*$. The expert policy is derived from direct usage of ground-truth targets or less noisy version filtered by sequence distillation (Kim and Rush, 2016). The objective is to maximize the following expectation:

$$\underbrace{\mathbb{E}_{\substack{\boldsymbol{y}_{\mathrm{del}} \sim d_{\tilde{\pi}_{\mathrm{del}}} \\ \boldsymbol{d}^* \sim \pi^*}} \sum_{d_i^* \in \boldsymbol{d}^*} \log \pi_\theta^{\mathrm{del}}(d_i^*|i, \boldsymbol{y}_{\mathrm{del}})}_{\textit{Deletion Objective}} + \underbrace{\mathbb{E}_{\substack{\boldsymbol{y}_{\mathrm{ins}} \sim d_{\tilde{\pi}_{\mathrm{ins}}} \\ \boldsymbol{p}^*, \boldsymbol{t}^* \sim \pi^*}} \left[ \sum_{p_i^* \in \boldsymbol{p}^*} \log \pi_\theta^{\mathrm{plh}}(p_i^*|i, \boldsymbol{y}_{\mathrm{ins}}) + \sum_{t_i^* \in \boldsymbol{t}^*} \log \pi_\theta^{\mathrm{tok}}(t_i^*|i, \boldsymbol{y}_{\mathrm{ins}}') \right]}_{\textit{Insertion Objective}},$$

where $\boldsymbol{y}_{\mathrm{ins}}'$ is the output after inserting palceholders $\boldsymbol{p}^*$ upon $\boldsymbol{y}_{\mathrm{ins}}$. $\tilde{\pi}_{\mathrm{del}}, \tilde{\pi}_{\mathrm{ins}}$ are the *roll-in* polices and we repeatedly draw states (sequences) from their induced state distribution $d_{\tilde{\pi}_{\mathrm{del}}}, d_{\tilde{\pi}_{\mathrm{ins}}}$. These states are first executed by the *expert* policy returning the suggested actions by the expert, and then we maximize the conditional log-likelihood over them. By definition, the *roll-in* policy determines the state distribution fed to $\pi_\theta$ during training. In this work, we have two strategies to construct the roll-in policy — adding noise to the ground-truth or using the output from the adversary policy. Figure 2 shows a diagram of this learning paradigm. We formally write down the roll-in policies as follows.

1. *Learning to Delete*: we design the $\tilde{\pi}_{\mathrm{del}}$ as a stochastic mixture between the initial input $\boldsymbol{y}^0$ or the output by applying insertion from the model with some mixture factor $\alpha \in [0, 1]$:

$$d_{\tilde{\pi}_{\mathrm{del}}} = \{\boldsymbol{y}^0 \ \ \text{if} \ \ u < \alpha \ \ \text{else} \ \ \mathcal{E}\left(\mathcal{E}\left(\boldsymbol{y}', \boldsymbol{p}^*\right), \tilde{\boldsymbol{t}}\right), \ \ \boldsymbol{p}^* \sim \pi^*, \tilde{\boldsymbol{t}} \sim \pi_\theta\} \tag{6}$$

   where $u \sim \mathrm{Uniform}[0, 1]$ and $\boldsymbol{y}'$ is any sequence ready to insert tokens. $\tilde{\boldsymbol{t}}$ is obtained by sampling instead of doing argmax from Eq. (5).

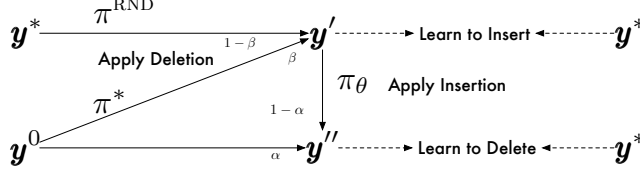

Figure 2: The data-flow of learning.

2. *Learning to Insert*: similar to the deletion step, we apply a mixture of the deletion output and a random word dropping sequence of the round-truth, inspired by recent advances of training masked language model (Devlin et al., 2018). We use random dropping as a form of noise injection to encourage more exploration. Let $\beta \in [0, 1]$ and $u \sim \text{Uniform}[0, 1]$,

$$d_{\tilde{\pi}_{\text{ins}}} = \{\mathcal{E}\left(\boldsymbol{y}^0, \boldsymbol{d}^*\right), \ \boldsymbol{d}^* \sim \pi^* \ \text{ if } \ u < \beta \ \text{ else } \ \mathcal{E}\left(\boldsymbol{y}^*, \tilde{\boldsymbol{d}}\right), \ \tilde{\boldsymbol{d}} \sim \pi^{\text{RND}}\} \qquad (7)$$

**Expert Policy**   It is crucial to construct an expert policy in imitation learning which cannot be too hard or too weak to learn from. Specifically, we considered two types of experts:

1. *Oracle*: One way is to build an oracle which accesses to the ground-truth sequence. It returns the optimal actions $\boldsymbol{a}^*$ (either oracle insertion $\boldsymbol{p}^*, \boldsymbol{t}^*$ or oracle deletion $\boldsymbol{d}^*$) by:

$$\boldsymbol{a}^* = \underset{\boldsymbol{a}}{\arg\min} \, \mathcal{D}(\boldsymbol{y}^*, \mathcal{E}(\boldsymbol{y}, \boldsymbol{a})) \qquad (8)$$

Here, we use the Levenshtein distance (Levenshtein, 1965)[2] as $\mathcal{D}$ considering it is possible to obtain the action suggestions efficiently by dynamic programming.

2. *Distillation*: We also explore to use another teacher model to provide expert policy, which is known as sequence-level knowledge distillation (Kim and Rush, 2016). This technique has been widely used in previous approaches for nonauoregressive generation (Gu et al., 2018). More precisely, we first train an autoregressive teacher model using the same datasets and then replace the ground-truth sequence $\boldsymbol{y}^*$ by the beam-search result of this teacher-model, $\boldsymbol{y}^{\text{AR}}$. We use the same mechanism to find the suggested option as using the ground-truth oracle.

### 3.3   Inference

**Greedy Decoding**   At inference time, we apply the trained model over the initial sequence $\boldsymbol{y}^0$ for several iterations. We greedily pick up the actions associated with high probabilities in Eq. (3)(4)(5). Moreover, we find that using search (instead of greedy decoding) or nosiy parallel decoding (Cho, 2016) does not yield much gain in LevT. This observation is quite opposite to what has been widely discovered in autoregressive decoding. We hypothesize there may be two reasons leading to this issue: (i) The local optimal point brought by greedy decoding in autoregressive models is often far from the optimal point globally. Search techniques resolve this issue with tabularization. In our case, however, because LevT inserts or deletes tokens dynamically, it could easily revoke the tokens that are found sub-optimal and re-insert better ones; (ii) the log-probability of LevT is not a good metric to select the best output. However, we do believe to see more improvements if we include an external re-ranker, e.g. an autoregressive teacher model. We leave this discussion in the future work.

**Termination Condition**   The decoding stops when one of the following two conditions is fulfilled:

1. *Looping*: Generation is terminated if two consecutive refinement iterations return the same output which can be (i) there are no words to delete or insert; (ii) the agent gets stuck in an infinite loop: i.e. the insertion and deletion counter each other and keep looping.

2. *Timeout*: We further set a maximum number of iterations (timeout) to guarantee a constant-time complexity in the worst case (Lee et al., 2018; Ghazvininejad et al., 2019).

**Penalty for Empty Placeholders**   Similar to Stern et al. (2019), we add a penalty to insert "empty" placeholder in decoding. Overly inserting "empty" placeholders may result in shorter output. A penalty term $\gamma \in [0, 3]$ is subtracted from the logits of 0 in Eq. (4).

Table 1: Generation quality (BLEU ↑, ROUGE-1/2/L ↑) and latency (ms ↓) as well as the average number of decoder iterations ($I_{\text{DEC}}$) on the standard test sets for LevT and the autoregressive baseline (with both greedy and beam-search outputs). We show the results of LevT trained from both oracle and the autoregressive teacher model.

|  | Dataset | Metric | Transformer | | Levenshtein Transformer | |
|  |  |  | greedy | beam4 | oracle | distillation |
|---|---|---|---|---|---|---|
| Quality ↑ | Ro-En | BLEU | 31.67 | 32.30 | 33.02 | **33.26** |
|  | En-De | BLEU | 26.89 | 27.17 | 25.20 | **27.27** |
|  | En-Ja | BLEU | 42.86 | **43.68** | 42.36 | 43.17 |
|  |  | ROUGE-1 | 37.31 | **37.87** | 36.14 | 37.40 |
|  | Gigaword | ROUGE-2 | 18.10 | **18.92** | 17.14 | 18.33 |
|  |  | ROUGE-L | 34.65 | **35.13** | 34.34 | 34.51 |
| Speed ↓ | Ro-En | Latency (ms) /$I_{\text{DEC}}$ | 326 / 27.1 | 349 / 27.1 | 97 / 2.19 | **90 / 2.03** |
|  | En-De | Latency (ms) /$I_{\text{DEC}}$ | 343 / 28.1 | 369 / 28.1 | 126 / 2.88 | **92 / 2.05** |
|  | En-Ja | Latency (ms) /$I_{\text{DEC}}$ | 261 / 22.6 | 306 / 22.6 | 112 / 2.61 | **106 / 1.97** |
|  | Gigaword | Latency (ms) /$I_{\text{DEC}}$ | 116 / 10.1 | 149 / 10.1 | 98 / 2.32 | **84 / 1.73** |

＿The ＿latter ＿coil ＿generated ＿2.2 T ＿in ＿liquid ＿helium . ⟶ ＿後者のコイルは 液体ヘリウム中で２．２Tを出した 。

| (iteration 1) | nothing to delete >> | |
|  | insert >> | [＿][後者][の][液体][液体][ヘリウム][ヘリウム][２．２][２．２][T][。] |
| (iteration 2) | delete >> | [＿][後者][の][液体][液体][ヘリウム][ヘリウム][~~２．２~~][２．２][T][。] |
|  | insert >> | [＿][後者][の][コイル][は][液体][ヘリウム][中で][２．２][T][発生した][。] |
| nothing to delete, nothing to insert >> | | **[Terminate]** |

Figure 3: An example of WAT'17 En-Ja translation with two decoder iterations by LevT. We present the inserted tokens in purple and deleted tokens with ~~red strikethrough~~.

# 4 Experiments

We validate the efficiency, effectiveness, and flexibility of Levenshtein Transformer extensively across three different tasks — machine translation (MT), text summarization (TS) and automatic post-editing (APE) for machine translation, from both generation (§4.1) and refinement (§4.2) perspectives.

## 4.1 Sequence Generation

For the sequence generation perspective, we evaluate LevT model on MT and TS. As a special case, sequence generation assumes empty $y^0 = $ <s></s> as input and no initial deletion is applied.

**Data & Evaluation**  We use three diversified language pairs for MT experiments: WMT'16 Romanian-English (Ro-En)[3], WMT'14 English-German (En-De)[4] and WAT2017 Small-NMT English-Japanese (En-Ja, Nakazawa et al., 2017)[5]. The TS experiments use preprocessed data from the Annotated English Gigaword (Gigaword, Rush et al., 2015)[6]. We learn byte-pair encoding (BPE, Sennrich et al., 2016) vocabulary on tokenized data. Detailed dataset statistics can be found in the Appendix. For evaluation metrics, we use BLEU (Papineni et al., 2002) for MT and ROUGE-1,2,L (Lin, 2004) for TS. Before computing the BLEU scores for Japanese output, we always segment Japanese words using KyTea [7].

**Models & Training**  We adopt the model architecture of Transformer base (Vaswani et al., 2017) for the proposed LevT model and the autoregressive baseline. All the Transformer-based models are

Table 2: Ablation study for Levenshtein Transformer on En-De (a) and Ro-En (b) translation tasks.

(a) Test BLEU for variant weight sharing. Baseline scores from Lee et al. (IT, 2018), Ghazvininejad et al. (MaskT, 2019) are included for reference.

(b) Test BLEU and deletion loss with variant roll-in polices.

| sharing | none | plh, ins | ins, del | all | IT | MaskT | roll-in | BLEU | NLL(del) |
|---|---|---|---|---|---|---|---|---|---|
| *oracle* | – | 25.50 | – | 25.20 | – | – | *Ours* | **33.02** | $\approx 0.202$ |
| *distill* | 25.11 | **27.73** | 24.90 | 27.27 | 21.61 | 26.56 | *DAE* | 31.78 | $\approx 0.037$ |

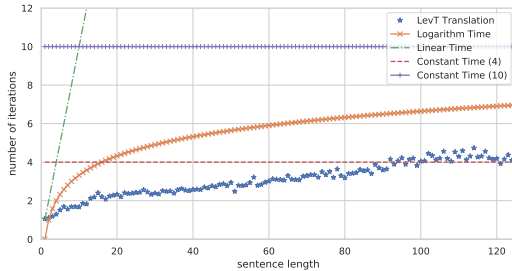

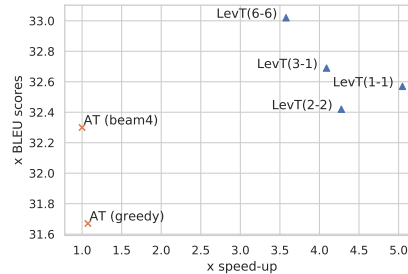

(a) Average number of refinement iterations v.s. length measured on monolingual corpus. For most of the time, LevT decodes with much smaller number (generally, 1∼4) of iterations.

(b) BLEU v.s. speed-up for LevT across variant early-exits and the autoregressive baselines on the test set of Ro-En.

Figure 4: Plots showing the decoding efficiency of the proposed Levenshtein Transformer.

trained on 8 Nvidia Volta GPUs with maximum $300K$ steps and a total batch-size of around $65,536$ tokens per step (We leave more details to the Appendix).

**Overall results** We present our main results on the generation quality and decoding speed in Table 1. We measure the speed by the averaged generation latency of generating one sequence at a time on single Nvidia V100 GPU. To remove the implementation bias, we also present the number of decoder iterations as a reference. It can be concluded that for both MT and summarization tasks, our proposed LevT achieves comparable and sometimes better generation quality compared to the strong autoregressive baseline, while LevT is much more efficient at decoding. A translation example is shown in Figure 3 and we leave more in Appendix. We conjecture that this is due to that the output of the teacher model possesses fewer modes and much less noisy than the real data. Consequently, LevT needs less number of iterations to converge to this expert policy.

**Ablation on Efficiency** As shown in Figure 4a, we plot the average number of iterations over the length of input over a monolingual corpus. LevT learns to properly adjust the decoding time accordingly. We also explore the variants of "early exit" where we denote LevT($m$-$n$) as a model with $m$ and $n$ blocks for deletion (Eq. (3)) and placeholder prediction (Eq. (4)) respectively. Figure 4b shows that although it compromises the quality a bit, our model with early exit achieves up to $\times 5$ speed-up (execution time) comparing against a strong autoregressive Transformer using beam-search.

**Ablation on Weight Sharing** We also evaluate LevT with different weight sharing as noted in §3.1. The results of models trained with oracle or distillation are listed in Table 2a. We observe that weight-sharing is beneficial especially between the two insertion operations (placeholder and token classifiers). Also, it shows another +0.5 BLEU improvement by not sharing the deletion operation with insertion compared to the default setting, which may indicate that insertion and deletion capture complementary information, requiring larger capacity by learning them separately.

**Importance of mixture roll-in policy** We perform an ablation study on the learning algorithm. Specifically, we train a model with no mixing of the $\pi_\theta$ in Equation (6). We name this experiment by DAE due to its resemblance to a denoising autoencoder. We follow closely a standard pipeline established by Lee et al. (2018). Table 2b shows this comparison. As we can see that the deletion loss

Table 3: Performance (BLEU ↑ / case-sensitive TER ↓) comparison on APE. "do nothing" represents the results of the original MT system output; the autoregressive model uses beam-size 4. For the proposed LevT, we use "scratch" to denote training from scratch on the APE triple data, and use "zero-shot" to denote applying an MT pre-trained LevT model directly for post-editing tasks. The same model can be further fine-tuned. All scores with underlines are from the model trained with an autoregressive teacher model (distillation) as the expert policy.

| Dataset | | MT system | Do-Nothing | Transformer | Levenshtein Transformer | | |
| --- | --- | --- | --- | --- | --- | --- | --- |
| | | | | | Scratch | Zero-shot | Fine-tune |
| Synthetic | Ro-En | PBMT | 27.5 / 52.6 | 28.9 / 52.8 | 29.1 / **50.4** | **30.1** / 51.7 | – |
| | | NMT | 26.2 / 56.5 | 26.9 / 55.6 | **28.3** / **53.6** | 28.0 / 55.8 | – |
| | En-De | PBMT | 15.4 / 69.4 | 22.8 / 61.0 | **25.8** / **56.6** | 16.5 / 69.6 | – |
| | En-Ja | NMT | 37.7 / 48.0 | 41.0 / 44.9 | **42.2** / **44.3** | 39.4 / 47.5 | – |
| Real | En-De | PBMT | 62.5 / 24.5 | 67.2 / 22.1 | 66.9 / 21.9 | 59.6 / 28.7 | **70.1 / 19.2** |

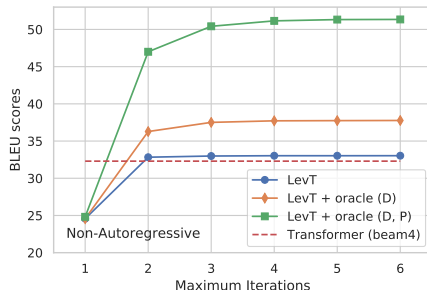

(a) Test set BLEU scores for WMT Ro-En

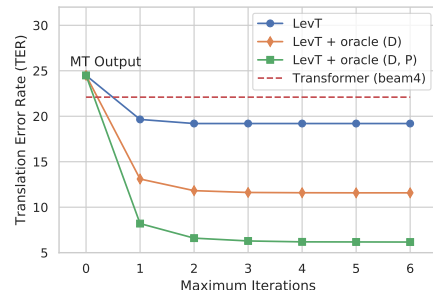

(b) Test set TER scores for Real APE En-De

Figure 5: MT & PE Performance v.s. Timeout iterations w/o oracle instructions.

from DAE is much smaller while the generation BLEU score is inferior. We conjecture that this is caused by the mismatch between the states from the model and the roll-in policy in training the DAE.

**v.s. Exiting Refinement-based Models**  Table 2a also includes results from two relevant recent works which also incorporate iterative refinement in non-autoregressive sequence generation. For fair comparison, we use the result with length beam 1 from Ghazvininejad et al. (2019). Although both approaches use similar "denosing" objectives to train the refinement process, our model explicitly learns "insertion" and "deletion" in a dual-policy learning fashion, and outperforms both models.

## 4.2 Sequence Refinement

We evaluate LevT's capability of refining sequence outputs on the APE task. In this setting, inputs are pairs of the source sequence and a black-box MT system generation. The ground-truth outputs are from real human edits with expansion using synthetic data.

**Dataset**  We follow a normal protocol in the synthetic APE experiments (Grangier and Auli, 2017): we first train the input MT system on half of the dataset. Then we will train a refinement model on the other half based on the output produced by the MT model trained in the previous phase. For the real APE tasks, we use the data from WMT17 Automatic Post-Editing Shared Task[8] on En-De. It contains both real PE triples and a large-scale synthetic corpus.

**Models & Evaluation**  The baseline model is a standard Transformer encoding the concatenation of the source and the MT system's output. For the MT system here, we want some imperfect systems that need to be refined. We consider a statistical phrase-based MT system (PBMT, Koehn et al., 2003) and an RNN-based NMT system (Bahdanau et al., 2015). Apart from BLEU scores, we additionally apply translation error rate (TER, Snover et al., 2006) as it is widely used in the APE literature.

**Overall results**   We show the major comparison in Table 3. When training from scratch, LevT consistently improves the performance of the input MT system (either PBMT or NMT). It also achieves better performance than the autoregressive Transformer in most of the cases.

**Pre-training on MT**   Thanks to the generality of the LevT model, we show it is feasible to directly apply the LevT model trained by generation onto refinement tasks — in this case — MT and APE. We name this a "zero-shot post-editing" setting. According to Table 3, the pre-trained MT models are always capable of improving the initial MT input in the synthetic tasks.

The real APE task, however, differs quite a bit from the synthetic tasks because human translators normally only fix a few spotted errors. This ends up with very high BLEU scores even for the "Do-nothing" column. However, the pre-trained MT model achieves the best results by fine-tuning on the PE data indicating that LevT is able to leverage the knowledge for generation and refinement.

**Collaborate with Oracle**   Thanks to the saperation of *insertion* and *deletion* operations, LevT has better interpretability and controllability. For example, we test the ability that LevT adapts oracle (e.g. human translators) instructions. As shown in Figure 5, both MT and PE tasks have huge improvement if every step the oracle *deletion* is given. This goes even further if the oracle provides both the correct *deletion* and the number of *placehoders* to insert. It also sheds some light upon computer-assisted text editing for human translators.

## 5   Related Work

**Non-Autoregressive and Non-Monotonic Decoding**   Breaking the autoregressive constraints and monotonic (left-to-right) decoding order in classic neural sequence generation systems has recently attracted much interest. Stern et al. (2018); Wang et al. (2018) designed partially parallel decoding schemes to output multiple tokens at each step. Gu et al. (2018) proposed a non-autoregressive framework using discrete latent variables, which was later adopted in Lee et al. (2018) as iterative refinement process. Ghazvininejad et al. (2019) introduced the masked language modeling objective from BERT (Devlin et al., 2018) to non-autoregressively predict and refine translations. Welleck et al. (2019); Stern et al. (2019); Gu et al. (2019) generate translations non-monotonically by adding words to the left or right of previous ones or by inserting words in arbitrary order to form a sequence.

**Editing-Based Models**   Several prior works have explored incorporating "editing" operations for sequence generation tasks. For instance, Novak et al. (2016) predict and apply token substitutions iteratively on phase-based MT system outputs using convolutional neural network. QuickEdit (Grangier and Auli, 2017) and deliberation network (Xia et al., 2017) both consist of two autoregressive decoders where the second decoder refines the translation generated by the first decoder. Guu et al. (2018) propose a neural editor which learned language modeling by first retrieving a prototype and then editing over that. Freitag et al. (2019) correct patterned errors in MT system outputs using transformer models trained on monolingual data. Additionally, the use of Levenshtein distance with dynamic programming as the oracle policy were also proposed in Sabour et al. (2018); Dong et al. (2019). Different from these work, the proposed model learns a non-autoregressive model which simultaneously inserts and deletes multiple tokens iteratively.

## 6   Conclusion

We propose Levenshtein Transformer, a neural sequence generation model based on insertion and deletion. The resulted model achieves performance and decoding efficiency, and embraces sequence generation to refinement in one model. The insertion and deletion operations are arguably more similar to how human writes or edits text. For future work, it is potential to extend this model to human-in-the-loop generation.

## Acknowledgement

We would like to thank Kyunghyun Cho, Marc'Aurelio Ranzato, Douwe Kiela, Qi Liu and our colleagues at Facebook AI Research for valuable feedback, discussions and technical assistance.

## Footnotes

[1]Codes for reproducing this paper are released in `https://github.com/pytorch/fairseq/tree/master/examples/nonautoregressive_translation`

[2]We only consider the variant which only computes insertion and deletion. No substitution is considered.

[3] http://www.statmt.org/wmt16/translation-task.html

[4] http://www.statmt.org/wmt14/translation-task.html

[5] http://lotus.kuee.kyoto-u.ac.jp/WAT/WAT2017/snmt/index.html

[6] https://github.com/harvardnlp/sent-summary

[7] http://www.phontron.com/kytea/

[8]http://www.statmt.org/wmt17/ape-task.html

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
