[Supplementary Material · Levenshtein_Transformer__Final_Version_ (1)-12-17.pdf]

# A  Learning & Inference Algorithm

We present the detailed algorithms for learning and decoding from Levenshtein Transformer as follows. For simplicity, we always omit the source information $x$ in conditional sequence generation tasks such as machine translation which is handled by the cross-attention with an encoder on $x$.

The learning algorithm is shown in Algorithm 1. $\mathcal{E}$ is the environment and $\mathcal{D}$ is denoted as the Levenshtein distance, and we can easily back-track the optimal insertion and deletion operations through dynamic programming. We only show the the case with single batch-size for convenience. We also present the inference algorithm in Algorithm 2. If the initial sequence $y^0$ is empty (`<s></s>`), the proposed model will skip the first deletion and do sequence generation. Otherwise, the model starts with deletion operations and refine the input sequence.

---

**Algorithm 1** Learning for Levenshtein Transformer

---

**Initialize:** Training set $\mathcal{T}$, expert policy $\pi^*$, model policy $\pi_\theta$, random deletion policy $\pi^{\text{RND}}$, $\alpha$, $\beta$
**repeat**
    Sample a training pair $(y^0, y^*) \sim \mathcal{Y}$
    **if** expert $\pi^*$ is a teacher model **then**
        Set the teacher's output as the target $y^* = y^{\text{AR}}$
    **end if**
    Sample $u, v \sim \text{Uniform}[0, 1]$
    **if** $u < \beta$ **then**
        $y_{\text{ins}} = \mathcal{E}(y^0, \tilde{d})$, where $\tilde{d} = \text{argmin}_d \mathcal{D}(y^*, \mathcal{E}(y^0, d))$
    **else**
        $y_{\text{ins}} = \mathcal{E}(y^*, \tilde{d})$, where $\tilde{d} \sim \pi^{\text{RND}}(\cdot | y^*)$
    **end if**
    $y'_{\text{ins}} = \mathcal{E}(y_{\text{ins}}, p^*)$, where $p^*, t^* = \text{argmin}_{p,t} \mathcal{D}(y^*, \mathcal{E}(y_{\text{ins}}, \{p, t\}))$
    **if** $v < \alpha$ **then**
        $y_{\text{del}} = y^0$
    **else**
        $y_{\text{del}} = \mathcal{E}(y'_{\text{ins}}, \hat{t})$, where $\hat{t} = \text{argmax}_t \sum_{y_i \in y'_{\text{ins}}, y_i = \text{<PLH>}} \log \pi_\theta^{\text{tok}}(t_i | i, y'_{\text{ins}})$
    **end if**
    $\mathcal{L}_\theta^{\text{ins}} = - \left[ \sum_{y_i \in y_{\text{ins}}, p_i^* \in p^*} \log \pi_\theta^{\text{plh}}(p_i^* | i, y_{\text{ins}}) + \sum_{y_i \in y'_{\text{ins}}, y_i = \text{<PLH>}, t_i^* \in t^*} \log \pi_\theta^{\text{tok}}(t_i^* | i, y'_{\text{ins}}) \right]$
    $\mathcal{L}_\theta^{\text{del}} = - \sum_{y_i \in y_{\text{del}}, d_i^* \in d^*} \log \pi_\theta^{\text{del}}(d_i^* | i, y_{\text{del}})$, where $d^* = \text{argmin}_d \mathcal{D}(y^*, \mathcal{E}(y_{\text{del}}, d))$
    $\theta = \theta - \lambda \cdot \nabla_\theta \left[ \mathcal{L}_\theta^{\text{ins}} + \mathcal{L}_\theta^{\text{del}} \right]$
**until** Maximum training steps reached

---

# B  Dataset and Preprocessing Details

Table 4 and 5 list the statistics (# of sentences, vocabulary) for all the datasets used in this work. We learn BPE vocabulary with $32,000$ joint operations for WMT En-De and Gigaword and $40,000$ joint operations for WMT Ro-En. For WAT En-Ja, we adopt the official $16,384$ BPE vocabularies learned separately on source and target side.

Table 4: Dataset statistics for sequence generation tasks (MT and TS).

|  | Dataset | Train | Valid | Test | Vocabulary |
|---|---|---|---|---|---|
| | WMT'16 Ro-En | 608,319 | 1999 | 1999 | 34,983 |
| Translation | WMT'14 En-De | 4,500,966 | 3000 | 3003 | 37,009 |
| | WAT'17  En-Ja | 2,000,000 | 1790 | 1812 | 17,952 / 17,801 |
| Summarization | English Gigaword | 3,803,957 | 189,651 | 1951 | 30,004 |

---

**Algorithm 2** Decoding for Levenshtein Transformer

---

**Initialize:** Input $\boldsymbol{y} = \boldsymbol{y}^0$, step $t = 0$, maximum step $T_{\max}$, model policy $\pi_\theta$.
**repeat**
  **if** $\boldsymbol{y} = \texttt{<s></s>}$ **then**
    Empty sequence, skip deletion: $\boldsymbol{y}' = \boldsymbol{y}$
  **else**
    Delete tokens: $\boldsymbol{y}' = \mathcal{E}(\boldsymbol{y}, \hat{\boldsymbol{d}})$, where $\hat{\boldsymbol{d}} = \arg\max_{\boldsymbol{d}} \sum_{y_i \in \boldsymbol{y}} \log \pi_\theta^{\mathrm{del}}(d_i|i, \boldsymbol{y})$
  **end if**
  **if** (t > 0) & ($\boldsymbol{y}' = \tilde{\boldsymbol{y}}$) **then**
    Termination condition satisfied: direct loop
    **break**
  **end if**
  Assign deleted output for back-up $\tilde{\boldsymbol{y}} = \boldsymbol{y}'$
  Insert placeholders: $\boldsymbol{y}'' = \mathcal{E}(\boldsymbol{y}', \hat{\boldsymbol{p}})$, where $\hat{\boldsymbol{p}} = \arg\max_{\boldsymbol{p}} \sum_{y_i y_{i+1} \in \boldsymbol{y}'} \log \pi_\theta^{\mathrm{plh}}(p_i|i, \boldsymbol{y}')$
  **if** $\boldsymbol{y}'' = \boldsymbol{y}' = \boldsymbol{y}$ **then**
    Termination condition satisfied: nothing to delete, nothing to insert.
    **break**
  **end if**
  **if** $\boldsymbol{y}'' = \boldsymbol{y}'$ **then**
    Nothing to insert, skip insertion: $\boldsymbol{y} = \boldsymbol{y}''$
  **else**
    Replace placeholders: $\boldsymbol{y} = \mathcal{E}(\boldsymbol{y}'', \hat{\boldsymbol{t}})$, where $\hat{\boldsymbol{t}} = \arg\max_{\boldsymbol{t}} \sum_{y_i \in \boldsymbol{y}'', y_i = \texttt{<PLH>}} \log \pi_\theta^{\mathrm{tok}}(t_i|i, \boldsymbol{y}'')$
  **end if**
  Update steps: t = t + 1
**until** Reach the maximum length $t = T_{\max}$
**return** $y$

---

Table 5: Dataset statistics for sequence refinement tasks (APE).

|  | Dataset | MT-Train | APE-Train | Valid | Test | Vocabulary |
|---|---|---|---|---|---|---|
| Synthetic | WMT'16 Ro-En | 300,000 | 308,319 | 1999 | 1999 | 34,983 |
|  | WMT'14 En-De | 2,250,000 | 2,250,967 | 3000 | 3003 | 37,009 |
|  | WAT'17 En-Ja | 1,000,000 | 1,000,000 | 1790 | 1812 | 17,952 / 17,801 |
| Real | WMT'17 APE En-De | 4,391,180 | 526,368 (fake) + 24,000 (real) | 2000 | 2000 | 40,349 |

## C   Model and Training Details

### C.1   Sequence Generation Tasks

Transformer models are used for autoregressive baselines as well as teacher models (for the expert policy). By default, we set $d_{\mathrm{model}} = 512$, $d_{\mathrm{hidden}} = 2048$, $n_{\mathrm{heads}} = 8$, $n_{\mathrm{layers}} = 6$, $\mathrm{lr}_{\mathrm{max}} = 0.0005$, label-smooth $= 0.1$, warmup $= 10000$ and dropout $= 0.3$. Source and target side share embeddings in all the training pairs except for WAT En-Ja where BPE vocabularies of both side are learned separately and are almost non-overlapping.

Since the training objectives for Levenshtein Transformer contains randomness terms (Eq. (6) (7)), we instead use BLEU (for MT) or ROUGE-2 (for TS) to select the best checkpoint by validation scores. We do not average checkpoints in this work.

### C.2   Sequence Refinement Tasks

For synthetic APE tasks, we keep the same training conditions for LevT as those for MT tasks (§C.1). As described earlier in §4.2, we build the baseline Transformer by concatenating the source and MT system's output as the input sequence for the encoder. Specially, we restart the positional embeddings

Table 6: The percentage of WMT En-De test sentence generation terminated at each iteration using LevT(T) with a maximum iteration of 10.

| Iterations | 1 | 2 | 3 | 4 | 5 | 6 | 7 | 8 | 9 | 10 | 2.43 |
|---|---|---|---|---|---|---|---|---|---|---|---|
| % | 12.3 | 48.1 | 28.5 | 8.5 | 2.0 | 0.4 | 0.1 | 0 | 0 | 0.1 | AVG |

for the MT output, add an additional language embedding for each token of the input sequence to show its language type. The detailed hyperpameters are the same as the standard Transformer.

As described in §4.2, we consider the following two different imperfect MT systems to provide the refinement inputs. Firstly, we consider the traditional statistical phrase-based machine translation system (PBMT). We follow the instruction to build the basic baseline model via moses[9]. As for the NMT-based model, we use a single layer attention-based model composed by LSTM. We build this model on fairseq-py[10] with the default configuration.

For the real APE task, we follow the procedures introduced in Junczys-Dowmunt and Grundkiewicz (2016). Synthetic corpus has two subsets: a 500K one and a 4M one. We over-sample real data by 10 times and merge it with the 500K synthetic data to train APE models. Besides, we also train a LevT MT model on the bigger (4M) synthetic corpus where we only use the source and target pairs.

### C.3 Implementation

Both the proposed Levenshtein Transformer and the baseline Transformer are implemented using PyTorch[11]. The codes are released as part of the Fairseq-py [12].

### C.4 Maximum Number of Iterations

We also presented in general how many sentences will be generated using the maximum iteration (for instance 10). As shown in Table 6, surprisingly, most predictions are gotten in 1-4 iterations, and the average number of iterations is 2.43. Only a tiny portion ($\sim 0.1\%$) require the maximum number of iterations demonstrating the efficiency of the proposed approach.

# D  More Decoding Examples

We present more examples from the proposed Levenshtein Transformer as follows.

Figure 6: Translation examples for WAT'17 Small-NMT En-Ja with the Levenshtein Transformer.

Figure 7: Translation examples for WMT'16 Ro-En with the Levenshtein Transformer.

**Figure 8 (a):**

| Or search for plan@@ ets similar to the Earth and thus perhaps discover ex@@ tr@@ ater@@ restri@@ al life ? → | Oder suchen Sie nach Planeten ähnlich der Erde und entdecken Sie damit vielleicht das ex@@ tr@@ ater@@ res@@ tr@@ ische Leben ? |
|---|---|

**(iteration 1)**

- *nothing to delete >>*
- *insert >>* — Oder suchen Sie Planeten Planeten Planeten Erde Erde Erde Erde und vielleicht Ex@@ tr@@ tr@@ tr@@ tr@@ ische entdecken ?

**(iteration 2)**

- *delete >>* — Oder suchen Sie ~~Planeten Planeten~~ Planeten ~~Erde Erde Erde~~ Erde und v~~ielleicht Ex@@ tr@@ tr@@ tr@@ tr@@ ische entdecken~~ ?
- *insert >>* — Oder suchen Sie nach Planeten ähnlich der Erde und entdecken so vielleicht außer@@ tr@@ dische Leben ?

**(iteration 3)**

- *delete >>* — Oder suchen Sie nach Planeten ähnlich der Erde und entdecken so vielleicht ~~außer@@ tr@@ dische~~ Leben ?
- *insert >>* — Oder suchen Sie nach Planeten ähnlich der Erde und entdecken so vielleicht das Leben ?

**(iteration 4)**

- *nothing to delete >>* — Oder suchen Sie nach Planeten ähnlich der Erde und entdecken so vielleicht das Leben ?
- *insert >>* — Oder suchen Sie nach Planeten ähnlich der Erde und entdecken so vielleicht das eigene Leben ?

*nothing to delete, nothing to insert >>* **[Terminate]**

**Figure 8 (b):**

| Local public transport will also become more expensive . → | Der öffentliche Nah@@ verkehr werde auch te@@ urer . |
|---|---|

**(iteration 1)**

- *nothing to delete >>*
- *insert >>* — Auch der öffentliche Nah@@ verkehr werden te@@ urer .

**(iteration 2)**

- *delete >>* — Auch der öffentliche Nah@@ verkehr ~~werden~~ te@@ urer .
- *insert >>* — Auch der öffentliche Nah@@ verkehr wird te@@ urer .

*nothing to delete, nothing to insert >>* **[Terminate]**

Figure 8: Translation examples for WMT'14 En-De with the Levenshtein Transformer.

**Figure 9 (a):**

| strengthened border patrol has led to a ##–percent drop in arrests of undocumented migrants this year at the u.s.–mexico frontier , it was reported on wednesday . → | arrests of bor@@ der–@@ cross@@ ers drop |
|---|---|

**(iteration 1)**

- *nothing to delete >>*
- *insert >>* — border patrol leads arrests ## migrants at u.s.–mexico border

**(iteration 2)**

- *delete >>* — border patrol ~~leads~~ arrests ~~##~~ migrants at u.s.–mexico border
- *insert >>* — border patrol reduces arrests of migrants at u.s.–mexico border

*nothing to delete, nothing to insert >>* **[Terminate]**

**Figure 9 (b):**

| us lawyer ed f@@ agan said wednesday he will bring a multi–million dollar lawsuit in the united states against the polish government unless it takes concrete steps to repay a huge debt to holders of bonds issued before world war ii . → | us star lawyer ed f@@ agan to sue poland for unpaid bonds |
|---|---|

**(iteration 1)**

- *nothing to delete >>*
- *insert >>* — f@@ agan to multi–million sue poland poland over debt debt

**(iteration 2)**

- *delete >>* — f@@ agan to ~~multi–million~~ sue ~~poland~~ poland over ~~debt~~ debt
- *insert >>* — f@@ agan threatens to sue poland over debt repayment

*nothing to delete, nothing to insert >>* **[Terminate]**

Figure 9: Translation examples for English Gigaword with the Levenshtein Transformer.

| | |
|---|---|
| (a) In the tag , insert the ActionScript code to create the behavior . | → | Fügen Sie im Tag den ActionScript–@@ Codes ein , um das Verhalten zu erstellen . |

(a)

In the tag , insert the ActionScript code to create the behavior . → Fügen Sie im Tag den ActionScript–@@ Codes ein , um das Verhalten zu erstellen .

(iteration 1)
*delete >>* ~~Klicken~~ Sie im ~~Tag des~~ ActionScript–@@ Codes ~~einfügen~~ , um das Verhalten zu erstellen .
*insert >>* Fügen Sie im Tag des ActionScript–@@ Codes ein , um das Verhalten zu erstellen .

*nothing to delete, nothing to insert >>* **[Terminate]**

(b)

In the tag , insert the ActionScript code to create the behavior . → Verwenden Sie die Schaltfläche " Bearbeiten , " um eine neue Java@@ Script@@ Aktion zu ändern oder zu erstellen .

(iteration 1)
*delete >>* Verwenden Sie die Schaltfläche " Bearbeiten , " um eine neue Ja~~vaScrip~~t Aktion zu ändern oder erstellen .
*insert >>* Verwenden Sie die Schaltfläche " Bearbeiten , " um eine neue JavaScript–@@ Aktion zu ändern oder zu erstellen .

*nothing to delete, nothing to insert >>* **[Terminate]**

(c)

To resize the canvas , drag the frame corners . → Um die Größe der Leinwand zu verändern , ziehen Sie die Rahmen@@ e@@ cken .

(iteration 1)
*delete >>* Um die Größe der Leinwand , ziehen Sie ~~den Rahmen .~~
*insert >>* Um die Größe der Leinwand zu ändern , ziehen Sie die Rahmen@@ e@@ cken .

*nothing to delete, nothing to insert >>* **[Terminate]**

Figure 10: Post-editing examples for WMT'17-APE En-De with the Levenshtein Transformer.

administratia t@@ v@@ r a facut constant eforturi pentru diminuarea cheltuielilor cu personalul si productia tv . → the administration of t@@ v@@ r has made constant efforts to reduce personnel and tv production expenses .

**MT**

(iteration 1)
*nothing to delete >>*
*insert >>* the t@@ v@@ r administration has constantly constantly efforts to cut spending on personnel and tv production .

(iteration 2)
*delete >>* the t@@ v@@ r administration has co~~nstantly co~~nstantly efforts to cut s~~pen~~ding on personnel and tv production .
*insert >>* the t@@ v@@ r administration has constantly made efforts to cut spending on personnel and tv production .

*nothing to delete, nothing to insert >>* **[Terminate]**

**APE (zero-shot on PBMT)**

(iteration 1)
*delete >>* the t@@ v@@ r ~~made~~ constant efforts to reduce expenditure on staff and tv production .
*insert >>* the t@@ v@@ r administration administration has making constant efforts to reduce expenditure on staff and tv production .

(iteration 2)
*delete >>* the t@@ v@@ r ~~administration~~ administration has ~~making~~ constant efforts to reduce expenditure on staff and tv production .
*insert >>* the t@@ v@@ r administration has made constant efforts to reduce expenditure on staff and tv production .

*nothing to delete, nothing to insert >>* **[Terminate]**

Figure 11: An example for machine translation and zero-shot post-editing over a PBMT system's output on WMT'16 Ro-En with the Levenshtein Transformer (LevT) trained for MT. It is clear to find that, the pre-trained LevT can directly adapt to the PBMT's output and have a different refinement results compared to translate from scratch.

## Footnotes

[9]`http://www.statmt.org/moses/?n=Moses.Baseline`

[10]`https://github.com/pytorch/fairseq/blob/master/fairseq/models/lstm.py`

[11]`https://pytorch.org/`

[12]`https://github.com/pytorch/fairseq/tree/master/examples/nonautoregressive_translation`