[Reviews · NeurIPS 2019]

Reviewer 1



[update] Thanks for the revision and clarification! I revised my review accordingly. ========================= This submission introduces Levenshtein Transformer, a non-autoregressive model for text generation and post editing. Instead of generating tokens left-to-right, it repeats a ete-and-sert procedure. More specifically, starting from an initial string, it keeps deletes tokens from or insert new tokens into the outputs, until convergence is met. The model is trained with imitation learning, where expert policy derived from gold data or a pertained auto-regressive teacher model is explored. Experiments on text summarization, machine translation, and post editing shows that the proposed model outperforms the transformer baselines in both accuracy and efficiency. Overall I think this is an interesting work. Yet I do have some confusion in both the technical part and the experimental part.

Reviewer 2



Originality: It is an interesting work by casting the sequence generation task as two iterative tasks of insertion/deletion. I think the formulation is new that is coupled with the training procedure based on imitation learning with two policies, i.e., deletion and insertion. Quality: The proposed model and its training procedure seem apt and well designed. Experiments are carried out carefully with consistent gains when compared with SOTA, i.e., Transformer, with faster inference speed. Clarity: This paper is clearly written, though I have a couple of minor questions regarding technical details. See the details in "Improvements" section. Significance: Given the inference efficiency and its reasonable quality improvements, I feel this work might have potential to impact future research. Other comment: line 89: we our policy for one iteration is -> {our, the}(?) policy for ...

Reviewer 3



=== Detailed Comments === > "two atomic operations — insertion and deletion" This is somewhat debatable. Under the LevT, an insertion operation first requires the number of slots to be predicted first, then the actual insertions are predicted. This is not completely atomic. i.e., using the authors terminology from Figure 1, "Insert Placeholders" then "Fill-in Tokens". > Section 1. "(up to ×5 speed-up" > Figure 4. > Section 4.1 "Analysis of Efficiency" This reviewer thinks the paper is quite misleading in the speed comparison, and iteration comparison. Figure 4 add/subtracts a U[0, 0.5) noise to the figure, which means it can subtract iterations -- this gives a misleading plot. Figure 4, and other iteration analysis is also misleading because the authors fail to take into account that 1 LevT iteration is 3 times more expensive than a standard transformer iteration (i.e., compared to other published methods). > Section 3.1 Roll-in Policy It took me several parses to fully understand the roll-in policy. This section can be rewritten to be more clear and easier to understand. > Section 3.2 Expert Policy and Section 4 "Oracle vs. Teacher Model" The terminology is confusing -- please use the standard terminology in the field -- this is simply Distillation vs no-Distillation. The Oracle and Teacher Model terminology is confusing. Additionally, the use of Levenshtein edit distance (and more specifically, decomposing it with dynamic programming, and using it as the oracle policy) is not new. Citations are missing [1, 2]. > Section 3.3 Comment: It seems like your model might benefit from a noisy-decoding approach. i.e., greedy decode with some noise, and select best one based off of log-prob of entire sequence. > Section 4 Machine Translation The authors presented several MT results, Ro-En, En-De, En-Ja -- this reviewer will focus on the WMT14 ende results. This is because WMT14 en-de is a well established benchmark for machine translation, and the other datasets are much less well established and lack strong prior work -- i.e., the other datasets are more interesting towards the NLP community and less so for the Machine Learning community. First, the reviewer thank the authors for published WMT14 ende, instead of taking the easy way out and only publishing on less competitive MT datasets. However, the empirical results are misleading. i.e., Table 1. The authors fail to compare to other prior published work while making bold claims on their own work. The Transformer baseline is very poor (26.X BLEU) -- it is behind the original Transformer paper [1] of 27.3 BLEU -- which in return is behind modern Transformer implementations which can reach >28 BLEU quite easily. > Section 4 Gigagword Similar to the MT results, citation and comparison to other published work is missing. For example, citing a few of the SOTA prior work from [4] would be nice. Overall, this reviewer argues for acceptance this paper. The ideas in the paper are sufficiently novel and is a good contribution to the community. The empirical results are lacking, but that should not be grounds for rejection. However, this reviewer also find the paper to be quite misleading in several places, especially in comparison with prior work and with a few citations missing. The writing and exposition of the paper can be improved. There are also many minor grammatical errors in the text, the text feels very rushed and definitely needs significant polishing. The exposition of the paper is definitely below the NeurIPS bar. However, assuming these issues are addressed in the rebuttal, this reviewer believes this paper should be accepted and will be willing to bump up the score from 6->7. [1] Optimal Completion Distillation for Sequence Learning [2] EditNTS: An Neural Programmer-Interpreter Model for Sentence Simplification through Explicit Editing [3] Attention Is All You Need [4] http://nlpprogress.com/english/summarization.html

[Author Response · NeurIPS 2019]

| AT(G) | AT(B) | LevT(O) | LevT(T) | It | 1 | 2 | 3 | 4 | 5 | 6 | 7 | 8 | 9 | 10 | 2.43 |
|---|---|---|---|---|---|---|---|---|---|---|---|---|---|---|---|
| 26.89 | 27.60 | 25.18 | 27.03 | % | 12.3 | 48.1 | 28.5 | 8.5 | 2.0 | 0.4 | 0.1 | 0 | 0 | 0.1 | AVG |

Table 1: (**Left**) Comparison of BLEU scores on the standard test set with Fairseq-py re-implementation; (**Right**) The percentage of test sentence generation terminated at each iteration using LevT(T) with a maximum iteration of 10.

We thank all the reviewers' insightful suggestions. We have revised the paper accordingly on language clarity and missing citations. For notations, we denote the standard Transformer as AT, and Levenshtein Transformer as LevT.

**Confusion in Empirical Results (R1, R3)**: We have migrated our codebase to Fairseq, a popular sequence-to-sequence learning framework. We hope this provides more trusty numbers for comparisons between LevT models and baseline AT models. The updated results on WMT14 En-De are presented in Table 1, where G, B, O and T are short for greedy decoding, beam search (with beam size 4), oracle and teacher model, respectively. Fairseq implementation basically achieves improved results on both LevT and the baseline models, but has larger gaps between the two. As it stands, this still holds our conclusion on both performance and speed-up. We will update all other tables with the new implementation in the final version.

For summarization (Gigaword) experiments, we attribute the small gap between our AT baseline and the one R1 referred to (35.19/17.58/32.98 v.s. 35.51/17.35/33.17 for ROUGE-1/2/L) to implementation difference (e.g. OpenNMT has copy attention while ours does not). We will add SoTA numbers in the final version as R3 suggested. And we point out that our models lack task-specific architectures (e.g. pointer-generator decoder [5], which is effective and has become popular) and direct comparison with SoTA models is not fair.

**Confusions on Iteration/Speed Comparison (R1, R3)**: We agree that injected noise in Figure 4 (a) is confusing, although it was originally for better visualization. We will remove it in the final version. We point out that our proxy for speed evaluation is the actual **machine execution time** [2, 4, 3], but not the number of iterations. We include the latter to show LevT's adaptive numbers of decoding iterations. And we clarify that a LevT iteration is not necessarily 3 times more expensive than an AT iteration (due to the "early exit" mechanism). We've updated Figure 4 (a) using the new implementation, and results on WMT14 En-De test set are presented in Table 1 (right). Most predictions are gotten in 1-4 iterations, and the mean is 2.43. Only a small portion ($\sim 0.1\%$) require the maximum number of iterations. From Figure 6 (in the Appendix), we see that the average number of iterations grows slowly with the sentence length.

**Confusions on Model Architecture (R1, R2)** A LevT decoding iteration contains one encoder forward pass and three decoder forward passes (for deletion, placeholder insertion and word filling). For the latter, we may not need to go through all the decoder layers, as described in Section 3.1 as "Early exit". In contrast, previous models require full passes of all the decoder layers [4].

**Why learning from teacher is better than oracle? (R1, R3)**: First, we thank the R3 for pointing out the terminology issue. We will certainly stick with the community on the normal terms. As observed by many prior work [2, 4, 3], distillation from a teacher model reduces the complex modalities our dataset possesses, which is shown critical for any non-autoregressive (NAT) based models including the proposed LevT.

**Imitation learning for baselines (R1)**: Teacher forcing is used as a standard technique to train AT models. From a high level standpoint, learning LevT falls into the same scope where the roll-out policy is from an expert, but the roll-in policy is a mixture. This is due to (i) lack of ground-truth path for target generation; (ii) the complementary insertion/deletion learning from the counterpart policy. Standard AT model already has the gold path provided by teacher forcing, so it is not necessary (and not possible) to apply the same algorithm to learn AT model.

**Noisy Parallel Decoding (R3)** As mentioned in Sec 3.3, our initial trial of using beam-search inside each prediction does not brings up much gain ($\approx +0.2$ BLEU). We conjecture similar results will happen in noisy decoding [1] due to its similar nature to beam search. We hypothesize this is because (i) LevT is able to revoke wrong predictions by deletion; (ii) the log-prob of LevT is not a good measure to select the best output. However, we do believe to see more improvements if we include an external re-ranker (e.g. AT teacher [2], language model). We will include this discussion and experiments in the final version.

# References

[1] K. Cho. Noisy parallel approximate decoding for conditional recurrent language model. *arXiv preprint arXiv:1605.03835*.

[2] J. Gu, J. Bradbury, C. Xiong, V. O. Li, and R. Socher. Non-autoregressive neural machine translation. *ICLR2018*.

[3] Ł. Kaiser, A. Roy, A. Vaswani, N. Parmar, S. Bengio, J. Uszkoreit, and N. Shazeer. Fast decoding in sequence models using discrete latent variables. *ICML2018*.

[4] J. Lee, E. Mansimov, and K. Cho. Deterministic non-autoregressive neural sequence modeling by iterative refinement. *EMNLP2018*.

[5] A. See, P. J. Liu, and C. D. Manning. Get to the point: Summarization with pointer-generator networks. *ACL2017*.


[Meta-Review · NeurIPS 2019]

After considering the author feedback and other reviewers' comments, all of the reviewers agree that the paper is worthy of publication at NeurIPS. I believe that the paper does not quite convince us that the deletion operation is beneficial, or that Levenshtein Transformer has some benefits to offer in terms of translation quality in the context of recent papers such as iterative refinement (Lee et al., 2018), Constant time MT (Ghazvininejad et al, 2019), Insertion Transformer (Stern et al, 2019). I encourage the authors to try to answer these questions more directly if possible. Nevertheless, there are some novel aspects of the paper and the proposed technique that can help advance the field.